# Effect of Total Dose Irradiation on Parasitic BJT in 130 nm PDSOI MOSFETs

**DOI:** 10.3390/mi14091679

**Published:** 2023-08-28

**Authors:** Yupeng Jia, Zhengxuan Zhang, Dawei Bi, Zhiyuan Hu, Shichang Zou

**Affiliations:** 1Shanghai Institute of Microsystems and Information Technology, Chinese Academy of Sciences, Shanghai 200050, Chinazsc@mail.sim.ac.cn (S.Z.); 2University of Chinese Academy of Sciences, Beijing 100049, China

**Keywords:** total dose irradiation, parasitic BJT, partially depleted SOI

## Abstract

In this work, the effects of total dose irradiation on the parasitic bipolar junction transistor (BTJ) in 130 nm PDSOI MOSFETs were investigated. The experimental results demonstrate that irradiation-induced oxide-trap charges can modify the E-B junction barrier, and thereby make the common-emitter gain $\beta_{0}$ of the parasitic BJT in NMOS device increase, while decreasing it in a PMOS device. Additionally, irradiation-generated oxide-trap charges in shallow trench isolation (STI) elevate the surface electrostatic potential of the gate above the STI sidewall, thus providing an additional channel from the emitter to the collector. Moreover, these charges may generate parasitic reverse conductive paths at the STI/Si interface under high dose irradiation, thereby enhancing the leakage current in the front gate channel and diminishing the significance of the parasitic BJT. Under irradiation, the electric field intensity difference between two biases leads to higher $\beta_{0}$ of the parasitic BJT in PG-biased devices than in ON-biased ones. Furthermore, the lifting effect of irradiation on $\beta_{0}$ increases in wide or short channel irradiated devices, which can be explained using simulations and an emitter current crowding effect model.

## 1. Introduction

With the rapid advancement of space technology, electronic devices operating in the space environment are exposed to cosmic rays and charged particles. This exposure may result in the deterioration or even failure of device performance, consequently posing a severe threat to the stability and lifespan of spacecraft and satellites. Silicon-On-Insulator (SOI) technology, which isolates devices from substrate silicon using an insulating buried oxide (BOX) and achieves complete dielectric isolation of individual devices along with STI, provides comprehensive relief. In contrast to conventional bulk silicon devices, SOI technology fundamentally eliminates latch-up effects that occur in bulk silicon CMOS devices. The presence of BOX layer limits the collection volume of charges induced by irradiation to only the top silicon region, reducing the single particle flip cross-section, thus reducing the soft error sensitivity to a great extent. Although the SOI device has excellent resistance to single event effects (SEE) and transient irradiation [1,2], whereas in the meantime oxide-trap charges and interface-trap charges are generated in the oxide layer by the total ionizing dose (TID) effect [3]. The presence of BOX layer makes the TID effect in SOI structures more complicated, and the TID effect becomes the main cause of degradation in SOI device performance.

Compared with fully depleted SOI (FDSOI), partially depleted SOI (PDSOI) MOSFETs are commonly utilized in the anti-irradiation domain owing to the threshold voltage that is independent of the thickness of top silicon or the amount of charges introduced in the BOX layer [4]. However, floating body effects (FBE), such as the kink effect, anomalous subthreshold slope, and reduced drain breakdown voltage, occur due to the presence of a neutral region in PDSOI devices [5]. Previous studies have shown that when SOI device is subjected to single event upset (SEU) and transient irradiation, the the parasitic BJT might turn on and amplify the current generated by the irradiation due to FBE, resulting in a severe reduction in the resistance to irradiation [6]. Devices subjected to SEU and transient irradiation are frequently accompanied by TID effects. The oxide-trap charges generated in ${\mathrm{SiO}}_{2}$ and the interface-trap charges generated at the Si/${\mathrm{SiO}}_{2}$ interface by total dose irradiation have impacts on the electrical parameters such as the electric field distribution in the body region of the SOI MOSFETs, resulting in the modification of FBE. To mitigate these FBE, the most common approach is to use a body contact via layout techniques with an H-Gate or T-Gate configuration [7]. Nevertheless, although body contact suppresses these effects effectively, the ideal body contact cannot be achieved to avoided FBE completely due to the presence of resistance in the body region itself. Therefore it is essential to investigate the effects of total dose irradiation on the FBE of PDSOI devices, especially on the parasitic BJT effect. Thus we expect that it can provide a reference for future research on irradiation-resistant device structures and processes.

In this paper, we focus on the impacts of total dose irradiation on the parasitic BJT in PDSOI devices based on a 130 nm SOI commercial process with I/O NMOS and PMOS fabricated in typical dimensions. Devices with different structures, including gate shapes, dimensions, are investigated. Besides these, the influences of irradiation dose and bias conditions, have been also studied. This article is arranged as follows: Section 2 describes the experimental details, including the devices and methods utilized in the study. Section 3 presents the experimental results and combines them with TCAD simulations for further discussion. Finally, Section 4 provides the concluding remarks of this research.

## 2. Materials and Methods

### 2.1. Devices

The present study utilized H-Gate or T-Gate input/output (I/O) devices selected from a 130 nm PDSOI process platform that operates at a nominal voltage (VDD) of 3.3 V. All experiment samples were fabricated on 8-inch commercial UNIBOND wafers obtained from SOITEC Corporation in France, which employs a 0.13 μm commercial logic SOI process. After oxidative consumption during the process, the final thickness of the top silicon film, gate oxide layer, and BOX are about 85 nm, 6.4 nm, and 145 nm, respectively. The doping concentration is about $4.5\times 10^{17}{\mathrm{cm}}^{-3}$ of substrate, and is more than $10^{20}\;{\mathrm{cm}}^{-3}$ of source and drain region. Figure 1 shows that the device isolation was achieved using shallow trench isolation (STI) technology, with body contact established from both terminals of the gate used for H-Gate PDSOI device, whereas only one terminal was connected to the body for T-Gate PDSOI devices.

The irradiation experiments were conducted at the Xinjiang Institute of Physical and Chemical Technology, Chinese Academy of Sciences (Xinjiang, China). Devices were irradiated with ${}^{60}C_{o}$γ-rays as the radiation source, and the irradiation dose points are 30 krad(Si), 50 krad(Si) and 100 krad(Si), with a dose rate of 100 rad(Si)/s. Two bias conditions were employed during irradiation: on-state (ON) and transmission-gate (TG or PG), and the bias conditions are defined in Table 1. Electrical parameters were measured with a KEITHEY 4200-SCS test system (Tektronix Company, Cincinnati, OH, USA), and the device measurement was performed within 30 min after irradiation to avoid any annealing effect. All irradiation experiments and measurement process were performed at room temperature.

### 2.2. Methods

Figure 2 shows the schematic diagram of parasitic BJT in SOI NMOSFETs. The regions of emitter, base and collector in the N-P-N BJT are form of $n^{+}$ type source, p-type silicon body and $n^{+}$ type drain of the SOI device. In a PDSOI transistor, the neutral body region, which corresponds to the base region of the BJT, is suspended generally, resulting in the formation of a parasitic BJT with a floating base region. Although the body contact is implemented at the gate terminal on H-Gate or T-Gate device, it remains impossible to eliminate this floating body effect completely.

To characterize the properties of parasitic BJT, we conducted measurements of the collector current ($I_{c}$), base current ($I_{b}$) and emitter current ($I_{e}$) at varying levels of base voltage $V_{be}$. Additionally, a Gummel plot was constructed, utilizing the aforementioned currents as vertical coordinates and $V_{be}$ as horizontal coordinate. The common-emitter gain $\beta_{0}$ of parasitic BJT was also calculated by the following Formula (1):(1)$$\;\beta_{0}=\left(I_{c}-I_{ceo}\right)/I_{b}$$
where $I_{ceo}$ is the $I_{c}$ value when the base is open. To obtain an accurate characterization of the collector current $I_{c}$, it is crucial to ensure that there is minimal non-amplified collector current present. Accordingly, the $I_{c}$ is corrected by subtracting $I_{ceo}$ [8]. In order to keep the front gate channel of the device from being in inversion state and eliminate the field effect factor of $I_{c}$, the bias during the parasitic BJT gain measurement was in the state with the gate voltage of $V_{g}$ = −1 V, and collector (drain) voltage $V_{c}$ = 2 V.

For a parasitic NPN transistor in the SOI NMOSFETs, the currents of collector, base, and emitter are also shown in Figure 2.

The current at each terminal can be expressed as Equation (Equation 2):(2)$$\begin{array}{cc} I_{e} & =I_{ne}+I_{re}+I_{pe}\\ I_{c} & =I_{nc}+I_{co}+I_{ch}\\ I_{b} & =I_{pe}+I_{re}+\left(I_{ne}-I_{nc}\right)-I_{co} \end{array}$$
where $I_{ne}$, $I_{pe}$, and $I_{re}$ are the collector electron current, collector hole current, and recombination current of E-B junction, respectively; $I_{nc}$ and $I_{co}$ are the electron current and reverse current of B-C junction, respectively; $I_{ch}$ is the subthreshold current of the front gate channel; $I_{rb}$ = ($I_{ne}$−$I_{nc}$) is the recombination current of base region.

## 3. Experimental Results and Discussion

### 3.1. Effect of Radiation on I-V Characteristics

Figure 3 shows the characteristics of 10 μm/0.35 μm NMOS devices before and after irradiation. The transfer characteristic curves of H-Gate devices drift negatively after irradiation due to the positive charge generated in the gate oxide and BOX layer. It is worth noting that the threshold voltage shift under PG bias is more significant than that under ON bias for the same dose. Regarding T-Gate PDSOI NMOSFETs, the leakage current of the front gate transistor and the hump of the back gate transistor after irradiation are obvious due to the presence of STI sidewall parasitic transistors. The total dose effect resistance of the H-Gate devices under PG bias can reach 100 krad(Si), while the total dose resistance of T-Gate devices under ON bias is less than 30 krad(Si).

Table 2 presents the front and back gate threshold voltage drifts of H-Gate devices caused by TID under PG bias with different sizes. It can be clearly seen that, for the devices with the same channel width, the threshold voltage shift is larger as the channel length becomes smaller; while for a fixed device channel length, the degradation of threshold voltage increases significantly as the channel width increases. In other words, there is an enhanced radiation effect in short and wide channel transistor. The threshold voltage shift after irradiation can be expressed by the following formula:(3)$$\begin{array}{cc} \Delta V_{th} & =\frac{\Delta Q_{B}}{C_{ox}A}=\frac{Q_{B,post}-Q_{B,pre}}{C_{ox}WL} \end{array}$$
where $C_{ox}$ is the capacitance of oxide layer; $Q_{B}$ is the charges of depletion region controlled by gate; $Q_{B,pre}$ and $Q_{B,post}$ are the value of $Q_{B}$ before and after irradiation, respectively. For the device with shorter and wider channel, a larger threshold voltage drift means that irradiation may produce more net positive charges in the oxide, and this will be discussed in Section 3.2.3.

### 3.2. Effect of Total Irradiation on Parasitic BJT

#### 3.2.1. Total Dose Irradiation

Figure 4 depicts the alterations of the common-emitter gain $\beta_{0}$ on parasitic BJT in H-Gate PDSOI NMOS and PMOS devices, with dimensions of 10 μm/0.35 μm and 10 μm/0.3 μm, before and after irradiation, respectively. All the devices are under ON bias during irradiation. Notably, our observations reveal that $\beta_{0}$ increases in NMOS devices and decreases in PMOS devices subsequent to irradiation. These effects can be also seen from the $I_{c}$-$V_{ce}$ curves illustrated in Figure 5. The modifications to $\beta_{0}$ are attributed to variations in both the base current $I_{b}$ and collector current $I_{c}$, and the results are given in Figure 6. Specifically, the $I_{b}$ of parasitic BJT in both NMOS and PMOS devices exhibited a slight increase after irradiation, while the $I_{c}$ in NMOS device was observed to increase and suffers degradation in PMOS device following irradiation. For NMOSFETs at intermediate voltage, the two currents behave ideally increasing with $V_{be}$ at 61 mV/decade, which satisfy the general law of BJT [9]: $I\propto exp(qV_{be}/nkT)$, and the ideality factor, *n*, is 1.02. Then the base current exhibits *n* = 1.08 and the collector current displays *n* = 1.15 after irradiation. The increase of ideality factor after irradiation indicates the increase in surface recombination. Regarding irradiated PMOS device, the ideality factor of base current increases and it decreases for collector current.

From Equation (Equation 3), it is evident that the amount of variations in the base and collector currents following irradiation can be expressed as:(4)$$\begin{array}{cc} \Delta I_{b} & =\Delta I_{re}+\Delta I_{rb}\\ \Delta I_{c} & =\Delta I_{ne}-\Delta I_{rb} \end{array}$$

In the case of NMOS transistors, irradiation induces the generation of a substantial number of oxide-trap charges in the BOX layer of the device, concomitant with the formation of many interface-trap charges at the upper interface of the BOX. The presence of oxide-trap charges in the BOX region gives rise to negative mirror charges in the bulk region, which manifest as a reduced doping concentration in the p-type bulk region. This effect, as described by Equation (Equation 4), leads to an expansion of the depletion region of the E-B junction in the p-type base region, resulting in an increase in the emitter junction recombination current. In other words, $\Delta I_{re}$ > 0.

Furthermore, the interface-trap charges at the upper surface of the BOX layer act as an additional recombination center, which enhance the recombination efficiency of electrons and holes in the neutral base region. This enhancement results in an increase in recombination current in the base region, namely $\Delta I_{rb}$ > 0. Ultimately, it leads to an enhancement in base current after irradiation due to the increase in recombination current. Theoretically, the collector current should decrease due to the increased recombination current in the base region. However, with positive oxide-trap charges in the BOX layer causing an increment in the potential of the body region and reduction in the E-B junction barrier, the emitting efficiency of the emitter will be increased, i.e., $\Delta I_{ne}$ > 0. It can be concluded that the oxide-trap charges have a stronger effect on the collector current as compared to the decrease caused by interface-trap charges, thus $\Delta I_{c}$ > 0 [10]. As depicted in Figure 6a for a device with an aspect ratio of 10 μm/0.35 μm, the increase in collector current is more significant than that in base current, thus leading to the increase of $\beta_{0}$ after irradiation.

In PMOS transistors, the recombination current within the base region also experiences an increase due to the presence of interface-trap charges at the upper surface of the BOX layer. Furthermore, negative mirror charges increase the doping concentration in the n-type bulk region, resulting in a narrower depletion region, leading to a rise in the E-B junction barrier and a reduction in the efficiency of emitter emission. Ultimately, irradiation results in a decrease in collector current, which, in turn, leads to a decline in the amplification coefficient $\beta_{0}$.

Regarding T-Gate PDSOI NMOSFETs, Figure 7 shows the variations of $\beta_{0}$ in the device with an aspect ratio of 10 μm/0.35 μm before and after irradiation under ON bias conditions. It can be observed that the $\beta_{0}$ in NMOS devices increases after irradiation. The impacts of irradiation on $\beta_{0}$ arise from the variations of collector current $I_{c}$ and base current $I_{b}$, demonstrated in Figure 8 for irradiated devices. Analysis of the changes in $I_{b}$ and $I_{c}$ as function of $V_{be}$ reveals a slight increase in base current $I_{b}$ following irradiation, while the collector currents $I_{c}$ exhibits significant increase.

Following 30 krad(Si) irradiation, a minor shift in base current is observed. The collector current $I_{c}$ exhibits an increase as $V_{be}$ increases with a swing of approximately 200 mV/decade at a low $V_{be}$ region, gradually approaching to 135 mV/decade as $V_{be}$ increases further. The higher swing observed in the low $V_{be}$ region indicates the presence of current components beyond the intrinsic current of the parasitic BJT for collector current.

Despite the voltage of the front gate being 0 V, there remains a depletion region in the front gate channel caused by the heavy doping in the polysilicon gate in parasitic BJT MOS transistors. The depletion region width $x_{d}$ and front gate voltage $V_{fg}$ satisfy the following equations:(5)$$\begin{array}{cc} V_{fg} & =V_{FB}+\psi_{S}+\frac{\sqrt{2\varepsilon_{0}\varepsilon_{si}qN_{A}\psi_{S}}}{C_{OX}}\\ x_{d} & =\sqrt{\frac{2\varepsilon_{0}\varepsilon_{si}\psi_{S}}{qN_{A}}} \end{array}$$
where $V_{FB}$ is the flat-band voltage; $\psi_{S}$ is the surface potential, and $N_{A}$ is the doping concentration of the p-type semiconductor. When $\psi_{S}=2kT/\left(q\ln\left(N_{A}/n_{i}\right)\right)$, the front gate voltage measured is the threshold voltage at this time. For our experiment, the value of the threshold voltage is 0.65 V, and thereby the flat-band voltage is calculated to be −1 V approximately.

When the voltage of −1 V is applied to the front gate of fresh NMOS device, the depletion region in the front gate disappears and the collector current is solely attributed to the parasitic BJT. However, when it comes to T-Gate NMOS devices, the accumulation of oxide-trap charges on the STI sidewall after irradiation raises the electrostatic potential of the gate above the STI sidewall, resulting in the presence of depletion layer in the front gate. In quick succession, the depletion layer enlarges the regions of B-C junction and E-B junction. Thus it provides an additional channel from the emitter to the collector, causing an increase in collector current.

Additionally, the oxide-trap charges in the STI and BOX layers raise the body potential near them and lower the E-B junction barrier. Hence it increases the efficiency of the emitter and causes an increase in the collector current.

When the dose point reaches 50 krad(Si), the irradiation-induced increase in base current surpasses that seen at 30 krad(Si). For our device, the collector current $I_{c}$ abruptly increases to about $10^{-9}A$ when $V_{be}$ is 0 V. The main reason for the current shift is attributed to the gradual activation of the STI sidewall parasitic transistor as the device is exposed to 50 krad(Si) radiation. The generation of subthreshold current leads to leakage current of front gate channel. Under the condition of low $V_{be}$, i.e., low collector current density $j_{c}$, the parasitic BJT in the device is not turned on yet and the collector current is dominated by the front gate channel leakage current. As $V_{be}$ ($j_{c}$) increases to a certain value, the amplification of parasitic BJT allows it come to dominate the collector current, and it is manifested as the intrinsic amplification of parasitic BJT. However, as the base voltage $V_{be}$ ($j_{c}$) increases further, the amplification drops below 1 due to the high injection effect.

As the dose point increases up to 100 krad(Si), the STI sidewall parasitic transistor turns on almost entirely, resulting in further increases in the leakage current of the front gate channel. In this circumstance, the characteristics of the parasitic BJT become masked, and the intrinsic amplification of parasitic BJT is no longer shown for the irradiated device in the medium collector current density region. Hence, the device enters the high injection effect stage directly.

Given that the leakage current of front gate channel caused by the STI sidewall parasitic transistor in the T-Gate PDSOI NMOS transistor after irradiation, the intrinsic amplification of the parasitic BJT is no longer significant for irradiated device. Thereby, the following discussion will be about the H-Gate PDSOI device only.

#### 3.2.2. Irradiation Bias

Figure 9 presents the irradiation response of the parasitic BJT at various bias states. The common-emitter gain $\beta_{0}$ of parasitic BJT is about 5 before irradiation, and then it increases to 6, 7, and 10 after an irradiation dose point of 30, 50, and 100 krad(Si), respectively. These findings indicate that total dose irradiation has a minor impact on the increase of the parasitic BJT amplification under ON bias during irradiation. As for the PG bias, the common-emitter gain $\beta_{0}$ is about 9 following irradiation with 30 krad(Si). It subsequently increases to 23 at 50 krad(Si) irradiation, and finally increases to $10^{3}$ at 100 krad(Si) irradiation.

Previous studies [11,12,13] have indicated that PG bias is the worst bias of the BOX layer, while ON bias is the worst bias of the STI under irradiation. Due to the body contact at both terminals of body region [14] for the H-Gate NMOS transistor, the device becomes insensitive to oxide-trap charges induced by STI after exposure to radiation. Moreover, the figures suggest that irradiation has a more profound impact on enhancing the parasitic BJT common-emitter gain $\beta_{0}$ for PG bias, resulting from the stronger electric field inside the BOX layer of PG bias. Figure 10 presents the simulation results of the electric field intensity inside the device under PG bias and ON bias.

On the one hand, the more potent electric field inside the BOX causess a greater number of positive charges to be generated and captured in the BOX during irradiation. There will be a lot of electron-hole pairs produced in the oxide layer during total dose irradiation, and some of the holes are recombined with electrons swiftly. The proportion of uncombined holes is the hole yield $f_{y}$($E_{ox}$), which is reliant on the strength of the electric field $E_{ox}$ in the oxide layer. For ${}^{60}C_{o}$γ-rays, the hole yield $f_{y}$($E_{ox}$) increases with the electric field strength $E_{ox}$ [15]. The number of uncombined holes ($N_{h}$) in the oxide layer after irradiation can be expressed as [16]:(6)$$N_{h}=f\left(E_{ox}\right)g_{0}Dt_{ox}$$
where $g_{0}$ is the density of electron-hole pairs produced per unit dose; *D* is the total irradiation dose; and $t_{ox}$ is the thickness of the oxide layer.

The electric field propels the uncombined holes away from the gate, causing them to move gradually through polaron jumping. As the holes traverse, their movement is hindered as they are intermittently captured by hole traps located in the oxide layer, leading to the generation of positive oxide-trap charges.

On the other hand, the electric field in the BOX near the source/drain-body junction is stronger under PG bias, and causes a higher accumulation of oxide-trap charges at the source/drain-body junction. This phenomenon reduces the potential barrier of the E-B junction, resulting in a more noteworthy improvement in the emission efficiency.

#### 3.2.3. The Effect of Device Geometry

In this part, the effects of the the channel length and width on parasitic BJT in post-irradiation devices will be discussed. Figure 11 presents the radiation responses of the parasitic BJT in devices with dimensions of 10 μm/10 μm and 1.2 μm/0.35 μm. Compared to the aspect ratio 10 μm/0.35 μm device in the previous discussion, the results indicate that for devices with 10 μm channel width, the irradiation-induced boosting effect on $\beta_{0}$ drops dramatically as the channel length increases. It is notable that, in the case of the 10 μm/10 μm device, $\beta_{0}$ decreases as irradiation dose increases in the low current region. Conversely, for devices with fixed channel length, the boosting effect on $\beta_{0}$ due to irradiation reduces substantially as the channel width decreases.

Based on the above conclusion, the common-emitter gain $\beta_{0}$ exhibits an increasing trend following total dose irradiation, with larger $\beta_{0}$ values observed when irradiation dose rises. Thus, it can be inferred that, for devices with short and wide channel, there is an enhancement in the total dose effect of the parasitic BJT.

Figure 12 gives the common-emitter gain variations $\delta \beta_{0}$ of H-Gate PDSOI NMOSFETs with different dimensions. The devices are under PG bias during irradiation.

In short channel devices, the strong electric field in the BOX near the source-body and drain-body junction extends to the middle region of the body. Consequently, the electric field in the BOX near the middle region of the channel is stronger than that of the long channel device. Under PG bias, the sentaurus TCAD software was employed to simulate the electric field intensity at each location on the upper surface of the BOX along the channel length with different channel lengths. As depicted in Figure 13a, the horizontal coordinates are normalized by the device channel length. The simulation results indicate that the electric field intensity in the middle region of the device with a channel length of 0.35 μm is indeed greater than that of 1.2 μm. Thus, more charges accumulate in the BOX for short channel devices during the irradiation process because of the larger electric field strength. This observation is confirmed by the curves shown in Figure 13b, where the initial voltage degradation of oxide-trap charges ($\Delta V_{ot}$) and interface state charges ($\Delta V_{it}$) were extracted by the mid-gap charge separation method [17]. And the expression of threshold voltage shift is:(7)$$\Delta V_{th}=\Delta V_{ot}+\Delta V_{it}=\frac{q}{C_{ox}}\left(\Delta N_{ot}-\Delta N_{it}\right)$$
where $\Delta N_{ot}$ is the oxide-trap charges density, and $\Delta N_{it}$ is the interface-trap charge density.

Furthermore, the width of the neutral base region increases as the device channel length increases, resulting in an increase in the ratio of the base region recombination current $I_{rb}$. After irradiation, as an additional recombination center at the BOX upper interface, the interface-trap charges also have a significant effect on enhancing the base region current. Consequently, the corresponding increase in the collector current decreases. In the low current region, the base region recombination current $I_{rb}$ is the main component of the base region current, making the effect so prominent that the $\beta_{0}$ shows a decrease rather than an increase after irradiation, as shown in Figure 11a.

Figure 14a presents the simulation results of the electric field in the BOX along the channel width under PG bias irradiation. For devices with the same gate length of 0.35 μm, the gate width is 10 μm and 1.2 μm, respectively. The horizontal coordinate is normalized to the device channel width. The simulation results indicate that the electric field strength in the BOX of the wide channel device is significantly greater than that of a narrow channel device. This implies that more oxide-trap charges are generated and accumulated in the BOX of the wide channel device under irradiation, thereby leading to a more significant impact of irradiation on the amplification of the wide channel device. The result of charges separation is depicted in Figure 14.

The base current, which initially flows along the channel direction, must be shifted to the vertical channel direction due to the body contact connected from both terminals of the active region. Additionally, the resistance of base region contributes to part of the voltage drop, leading to a lower potential in the center of the base region than at the two terminals. As a result, the current density is lower in the base region than in the terminals. This manifests as the emitter current concentrating in the area near the body contact, which is similar to the emitter current crowding effect observed in traditional BJTs with vertical structures [18].

For an SOI transistor under PG bias, the source (drain) terminal is connected to high voltage, and the body region is led to ground through both terminals of the H-Gate device. As illustrated in Figure 15, there are two primary paths for the electric field lines between the source (drain) and the body contact [19], one through the lightly doped p-type body region, and the other through the BOX layer. For wider device, more electric field lines are directed from the source(drain) to the body contact region through the BOX layer, and the electric field lines in the BOX are denser. Hence, it results in a stronger electric field.

## 4. Conclusions

In this work, the effects of total dose irradiation on the parasitic BJT in 130 nm partially depleted SOI I/O device were investigated and analyzed. Our findings reveal that, for H-Gate PDSOI NMOS devices, the emitter emission efficiency and base region width of the parasitic BJT are altered by oxide-trap charges induced in the BOX layer following irradiation, whereas the interface-trap charges generated by irradiation improve the carrier recombination efficiency, and lead to variations in emitter and base current. Ultimately, charges captured in the BOX increase the common-emitter gain $\beta_{0}$ of parasitic BJT in NMOS devices and decrease it in PMOS devices. For T-Gate NMOS devices, the accumulation of oxide-trap charges in the STI region raises the potential in the gate and body regions, leading to an increase in amplification. However, as the irradiation dose increases, the sidewall parasitic transistor turns on and masks the characteristics of parasitic BJT.

Furthermore, we also discover that the influence of irradiation on $\beta_{0}$ is related to bias state and geometric effect. For H-Gate NMOS devices, the common-emitter gain $\beta_{0}$ under PG bias is higher than that under ON bias due to differences in the electric field intensity inside the device during irradiation under different biases. The 3D simulation is employed to analyze the impacts of device geometry. It is proved that for the devices with the same channel width, the electric field weakens as the channel length increases, resulting in a decreased impact of irradiation on $\beta_{0}$ enhancement. Additionally, the effect of interface-trap charges on parasitic BJT gain in long channel device is more significant than that of short channel device. Given the emitter current crowding effect of the parasitic BJT, the emitter current concentrates in the region near the body contact. Therefore, for a fixed device channel length, it follows that a narrower channel leads to a diminished lifting effect of irradiation on the common-emitter gain $\beta_{0}$.

## Figures and Tables

**Figure 1 micromachines-14-01679-f001:**
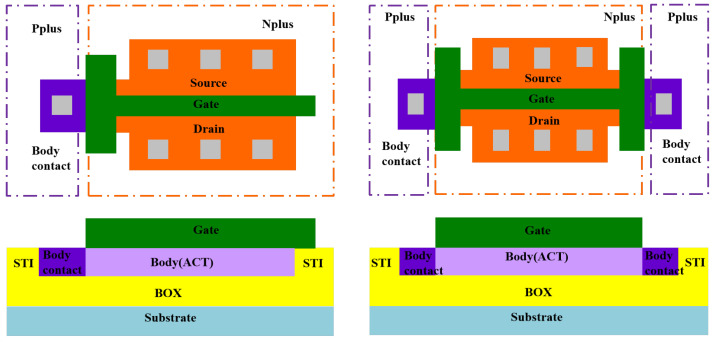
The structures of T-Gate and H-Gate PDSOI MOSFETs with external body contacts.

**Figure 2 micromachines-14-01679-f002:**
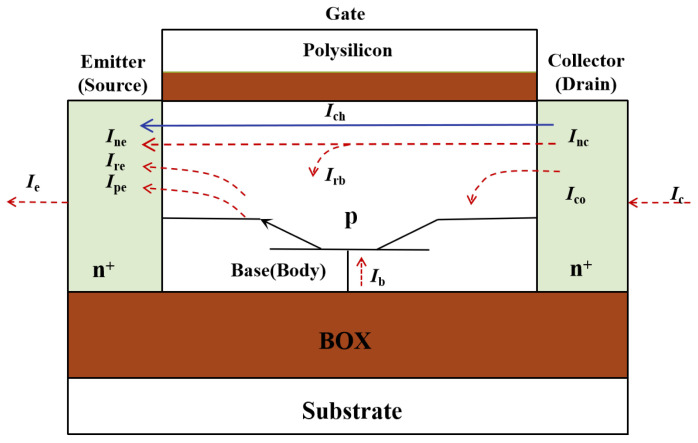
Schematic diagram of parasitic bipolar transistor effect in SOI NMOSFETs.

**Figure 3 micromachines-14-01679-f003:**
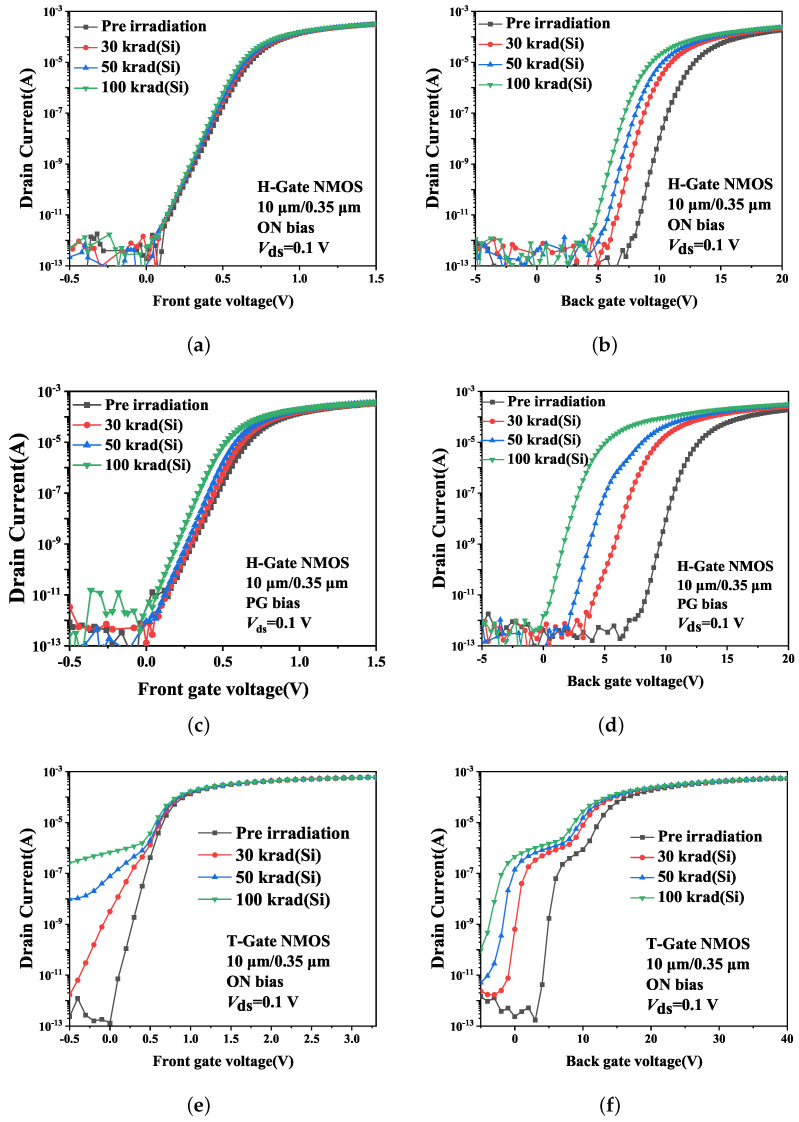
Transfer characteristics of front and back gate of 10 μm/0.35 μm NMOS devices after irradiation. (**a**,**c**,**e**) front gate; (**b**,**d**,**f**) back gate.

**Figure 4 micromachines-14-01679-f004:**
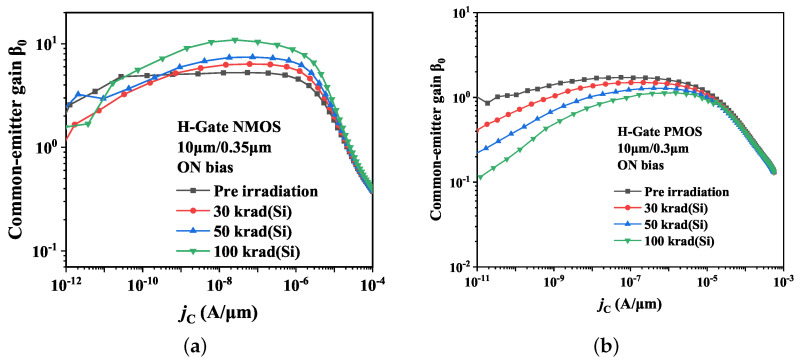
Effect of total dose irradiation on common-emitter gain $\beta_{0}$ of H-Gate I/O (**a**) NMOSFETs and (**b**) PMOSFETs.

**Figure 5 micromachines-14-01679-f005:**
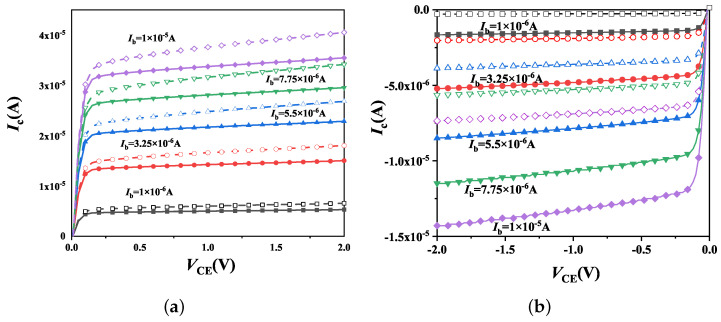
Effect of total dose irradiation on the parasitic BJT $I_{c}$-$V_{ce}$ curves of H-Gate I/O (**a**) NMOSFETs and (**b**) PMOSFETs. In the figures, solid lines represent un-irradiated devices and dashed lines represent devices irradiated with 100 krad(Si) under ON bias.

**Figure 6 micromachines-14-01679-f006:**
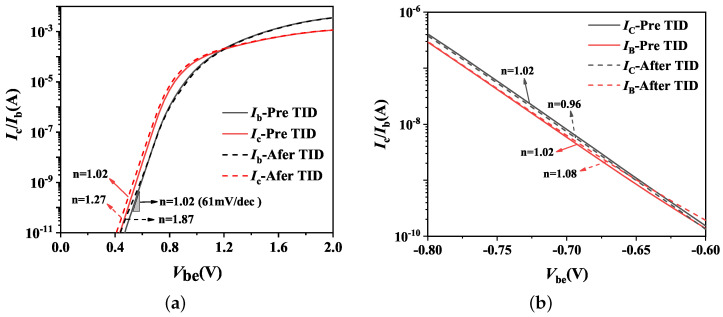
Effect of total dose irradiation on the parasitic BJT $I_{b}$ and $I_{c}$ of H-Gate I/O (**a**) NMOSFETs and (**b**) PMOSFETs.

**Figure 7 micromachines-14-01679-f007:**
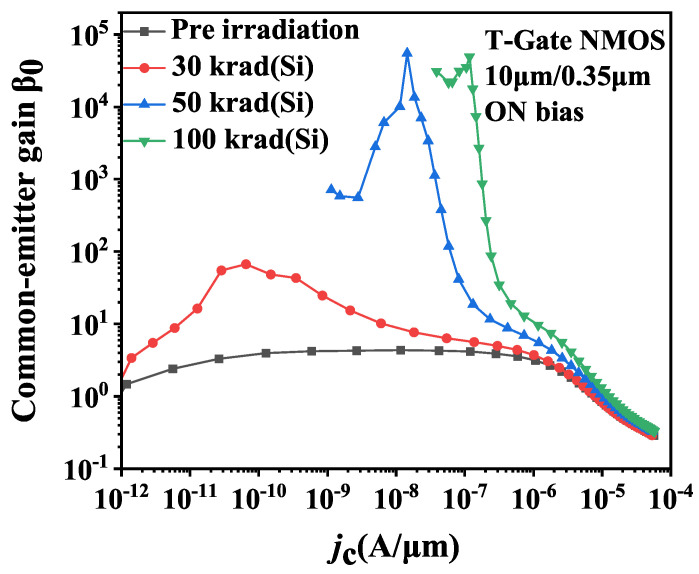
Effects of total dose irradiation on the common-emitter gain $\beta_{0}$ of T-Gate I/O NMOSFETs.

**Figure 8 micromachines-14-01679-f008:**
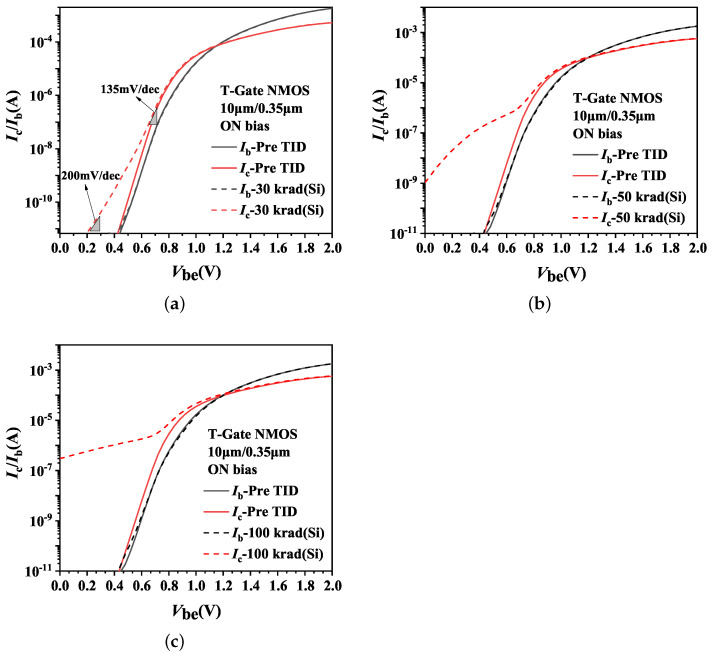
Effects of total dose irradiation on $I_{b}$ and $I_{c}$ of T-Gate I/O NMOSFETs. (**a**) 30 krad(Si); (**b**) 50 krad(Si); (**c**) 100 krad(Si).

**Figure 9 micromachines-14-01679-f009:**
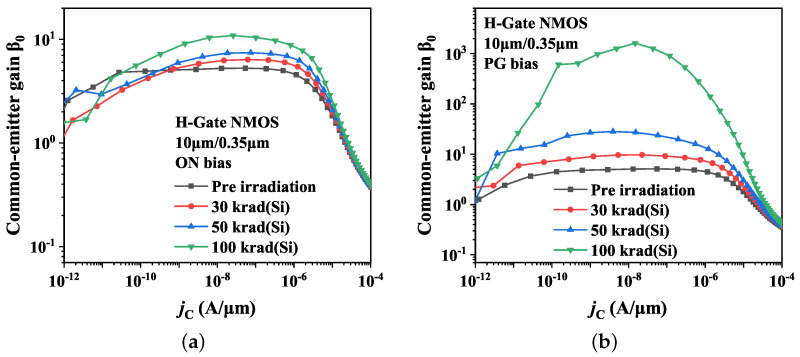
Effects of total dose irradiation on common-emitter gain $\beta_{0}$ of H-Gate I/O NMOSFETs at different bias states. (**a**) ON bias; (**b**) PG bias.

**Figure 10 micromachines-14-01679-f010:**
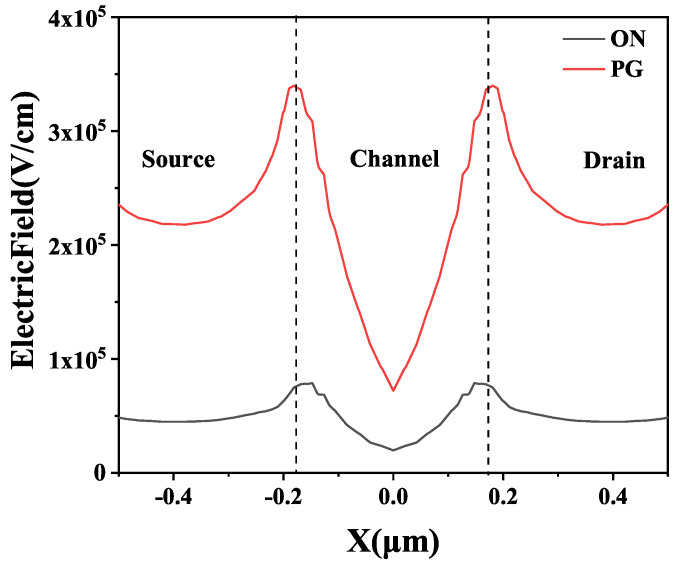
Distribution of electric field intensity along channel length at 2 nm under the interface between body region and BOX layers.

**Figure 11 micromachines-14-01679-f011:**
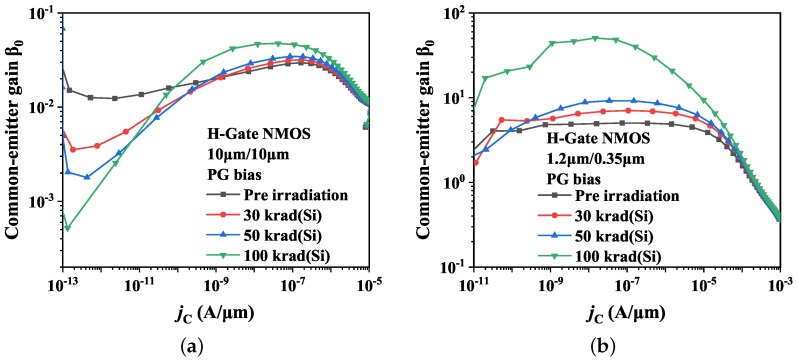
Effects of total dose irradiation on common-emitter gain $\beta_{0}$ of H-Gate I/O NMOSFETs with different dimensions. (**a**) 10 μm/10 μm; (**b**) 1.2 μm/0.35 μm.

**Figure 12 micromachines-14-01679-f012:**
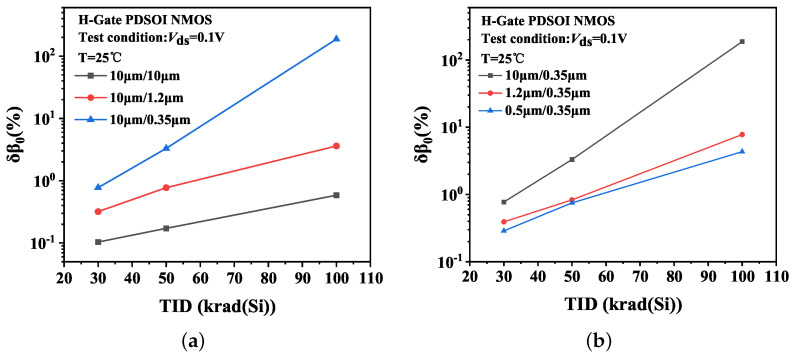
(**a**,**b**) Common-emitter gain variation $\delta \beta_{0}$ of H-Gate I/O NMOSFETs with different dimensions after irradiation.

**Figure 13 micromachines-14-01679-f013:**
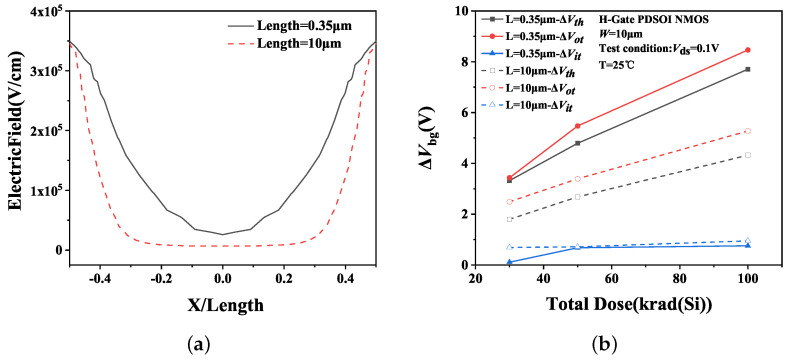
(**a**) Electric field intensity at each location along the channel length and (**b**) the $\Delta V_{th}$, $\Delta V_{ot}$ and $\Delta V_{it}$ of H-Gate I/O NMOSFETs after irradiation with different channel lengths.

**Figure 14 micromachines-14-01679-f014:**
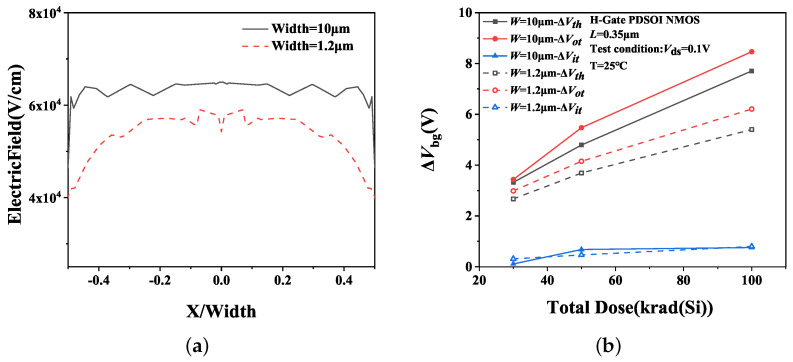
(**a**) Electric field intensity at each location along the channel width and (**b**) the $\Delta V_{th}$, $\Delta V_{ot}$ and $\Delta V_{it}$ of H-Gate I/O NMOSFETs after irradiation with different channel widths.

**Figure 15 micromachines-14-01679-f015:**
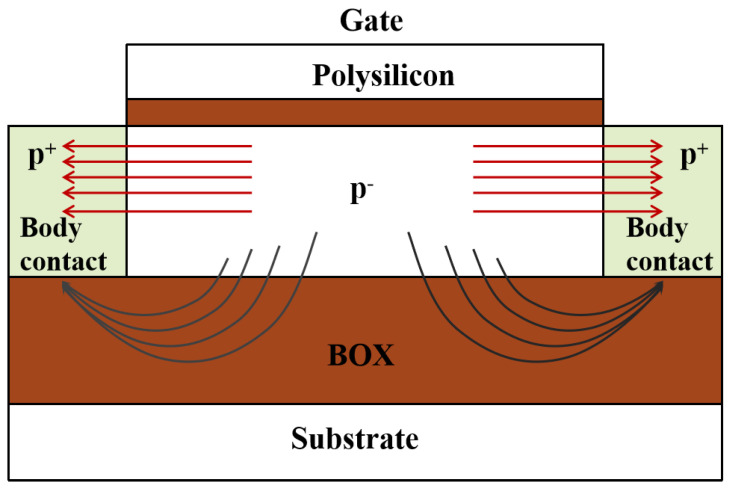
Schematic diagram of the internal electric field of the H-Gate device under PG bias.

**Table 1 micromachines-14-01679-t001:** The voltage of each terminal under different bias of the PDSOI transistor.

Type	Gate (V)	Source (V)	Drain (V)	Substrate (V)
ON	VDD	0	0	0
PG	0	VDD	VDD	0

**Table 2 micromachines-14-01679-t002:** The front and back gate threshold voltage drift caused by TID under PG bias of H-Gate NMOS devices with different sizes.

Parameter		$\Delta V_{t,fg}$ (mV)			$\Delta V_{t,bg}$ (V)	
**TID**	**30 krad(Si)**	**50 krad(Si)**	**100 krad(Si)**	**30 krad(Si)**	**50 krad(Si)**	**100 krad(Si)**
10 μm/10 μm	−2.23	−4.85	−12.23	−1.74	−2.59	−4.22
10 μm/1.2 μm	−11	−35.04	−41.56	−2.42	−3.17	−4.7
10 μm/0.35 μm	−41.18	−61.18	−134.7	−3.37	−5.12	−7.74
1.2 μm/10 μm	−40.35	−63.45	−95.87	−3.28	−4.05	−7.02
0.15 μm/10 μm	−33.38	−53.69	−91.54	−2.7	−3.72	−5.48

## Data Availability

Not applicable.

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
