# Peer review of "Effect of Total Dose Irradiation on Parasitic BJT in 130 nm PDSOI MOSFETs"

_micromachines, 2023, doi:10.3390/mi14091679_

Round 1

Reviewer 1 Report

The effects of total dose irradiation on the parasitic bipolar junction transistor (BTJ) in 130 nm PDSOI MOSFETs were investigated by using simulation and experiment in this manuscript. This work has innovative and practical value. However, the English language description is not very clear and there are many minor errors that need to be corrected. These points should be considered by the authors in making revisions to their paper, after which the paper could be accepted for publication.

Author Response

Dear reviewer:

We gratefully appreciate for your comment. I have carefully revised my manuscript and the English language description was also modified. Attachments is latest version. We sincerely hope that you find our responses and modifications satisfactory and that the manuscript is now acceptable for publication.

Reviewer 2 Report

Submitted paper presents comprehensive and detailed study of total dose effects of PDSOI MOSFETs. Investigating the effects of different conduction types, gate dimensions and irradiation conditions, degradation behavior is explained based on operation of parasitic bipolar transistors (BJTs) formed in the MOSFFETs. Although the radiation effects can be understood following author’s proposed BJT model, I think characteristics of the MOSFETs should be also shown to verify it. I-V characteristics and the threshold voltage shift of irradiated MOSFETs will be very helpful to understand the importance of this study, as authors previously reported in IEICE Electron. Express 17, 1-6.

Author Response

Dear reviewer:

We gratefully appreciate for your comment. I take your point that it is important to show the transfer characteristics of the MOSFETs. Therefor, a new section--Effect of Radiation on I-V Characteristics has been added to describe the characteristics of irradiated MOSFETs in the latest version. The I-V characteristic curves of the devices with different bias states, gate shapes have been shown, and the threshold voltage drifts of the devices with different sizes were also presented. In addition, the total dose effect model of the devices with different sizes were verified by the separation of oxide charges through the mid-band voltage method, and it also explained the impacts of device geometry on β0 variation of irradiated devices. The addition has been highlighted in the revised manuscript and the English language description was also modified. We sincerely hope that you find our responses and modifications satisfactory and that the manuscript is now acceptable for publication.

Round 2

Reviewer 2 Report

 Resubmitted paper presents more clearly irradiation effect on parasitic BJT in PDSOI MOSFET. However, I feel this manuscript still includes some points to be addressed and corrected. The following are comments;

1.  Please show source and drain contacts in fig. 1(a).

2.  Thickness of the front gate oxide is 6.4nm, meanwhile, how thick BOX layer?

3.   Please refer to channel width and length in 2.1 Devices.

4.  Page 6, line 125-127; “ For the device with shorter and wider channel,…in section 3.4.”

  Section 3.2.3 is correct?

5. Page 7, fig.5;

  Figures are shown oppositely. Figure 5 (b) should be for a nMOSFET (npn transistor).  According to a past report on Si bipolar junction transistors (BJTs) [*], oxide tapped and interface charges of passivation oxide over base-emitter junction is responsible for radiation degradation. Surface recombination will be increased due to those charges. Please estimate ideality factor from fig. 5(a) and (b) to evaluate effect of recombination current [**].

[*] D.M. Schmidt et al., IEEE Trans. Nucl. Sci. 42, 1541(1995).

[**] D. K. Schroder, Semiconductor material and device characterization, 3rd ed., John Wiley & Sons, Inc.: Hoboken, NJ, USA, 2006; p. 199.

6. From page 7, line 140 to page 10, line 223.

Please interpret degradation of transfer characteristics based on proposed model in [*] to verify the author’s model. Besides, please cite appropriate references or other means of supporting author’s model.

7. Page 11, fig.10 (a)

 Captions for each part or layer are needed.

8. Page 11,fig.10 (b)

Why electric field is symmetry under PG bias, regardless of VDD is applied to source electrode? Why the field at channel is smaller for ON bias, under which VDD is applied to the gate?

9. Page 12, fig. 11 (a),(b);

  Why gain becomes smaller with increasing channel length and decreasing its width?

10. Page 13, line 285-287;

  Estimation method should be described in detail citing references.

11. Page 12-14.

Although authors discuss the effect of device geometry in terms of electric field generated in BOX layer, I think the effect can be simply understand based on negatively increased midgap or flat band voltage, rather than parasitic BJT. Surface potential bending due to the voltage gives rise to increased electric field in the gate oxide and BOX layer. Densities of positive charges accumulated in them and interface traps formed at the gate and BOX/Si are increased. It is inherent dimensional effect, since drain current of MOSFET ideally proportional to W/L. Accordingly, estimated mid gap current and voltage are also affected by W/L.

Author Response

Dear reviewer:

We gratefully appreciate for your valuable suggestions. Due to my negligence, there are some small mistakes in it. Please forgive me for the inconvenience caused. In addition, I will reply to your questions and explain them one by one. And the modified parts have been  highlighted in the manuscript.

  1. Comment: Please show source and drain contacts in fig. 1(a).

Response:

For the T/H gate device used in our experiment, it belongs to the source-drain symmetrical structure. I have labeled Source and drain for the reader's convenience.

  1. Comment: Thickness of the front gate oxide is 6.4nm, meanwhile, how thick BOX layer?

Response:

For the PD SOI device in this manuscript, the thickness of the BOX is 145 nm.

  1. Comment: Please refer to channel width and length in 2.1 Devices.

Response:

The device diagram in section 2.1 is a structural diagram, and it is used to characterize the difference between the T-gate and H-gate device.

  1. Comment: Page 6, line 125-127; “ For the device with shorter and wider channel,…in section 3.4.”  Section 3.2.3 is correct?

Response:

Annotated chapter errors have been revised.

  1. Comment: Figures are shown oppositely. Figure 5 (b) should be for a nMOSFET (npn transistor).  According to a past report on Si bipolar junction transistors (BJTs) [*], oxide tapped and interface charges of passivation oxide over base-emitter junction is responsible for radiation degradation. Surface recombination will be increased due to those charges. Please estimate ideality factor from fig. 5(a) and (b) to evaluate effect of recombination current [**].

[*] D.M. Schmidt et al., IEEE Trans. Nucl. Sci. 42, 1541(1995).

[**] D. K. Schroder, Semiconductor material and device characterization, 3rd ed., John Wiley & Sons, Inc.: Hoboken, NJ, USA, 2006; p. 199.

Response:

Figures 5(a) and 5(b) have been revised to the correct position, and we have estimated the ideality factor of parasitic BJT before and after irradiation, as detailed in the latest version.

  1. Comment: Please interpret degradation of transfer characteristics based on proposed model in [*] to verify the author’s model. Besides, please cite appropriate references or other means of supporting author’s model.

Response:

It is proposed in [*] that the degradation of BJT gain after irradiation is due to the influence of the oxide-trap charge and interface-trap charge generated by irradiation in the oxide layer. For H-gate devices discussed in this paper, the charges generated by the gate oxide layer is almost negligible due to its thin layer. Therefore the effect of charges in BOX layer is mainly considered.

(1) The first mechanism: the oxide-trap charges enlarge the depletion region of E-B junction in the P-type base region, which increase the recombination current of the emitter junction.

(2) The second mechanism: the interface-trap charges, as an additional recombination center, increase the recombination current in the base region.

(3) The third mechanism: at the same time, the positive oxide-trap charges increase the potential of the body region, reduce the potential barrier of E-B junction, increase the emitter electron injection, and improve the emission efficiency of the emitter.

The (1)(2) mechanisms lead to increase in base current. The (3) mechanism causes an increase in collector current, and this influence is more significant than base current increase. This is consistent with the mechanism of gain degradation induced by irradiation proposed by [*].

  1. Comment: Page 11, fig.10 (a). Captions for each part or layer are needed.

Response:

The title of the picture has been revised.

  1. Comment: Why electric field is symmetry under PG bias, regardless of VDD is applied to source electrode? Why the field at channel is smaller for ON bias, under which VDD is applied to the gate?

Response:

The PG bias state : electrode voltage has been modified. The PG bias is a VDD voltage applied to both the source and drain electrode. And PG bias creates a larger electric field in the BOX than ON bias due to the electrode with bias added closer to the BOX .

  1. Comment: Page 12, fig. 11 (a),(b): Why gain becomes smaller with increasing channel length and decreasing its width?

Response:

This is an intuitive conclusion obtained by comparing the gain of three devices of different sizes in Fig. 9(b)10 µm/0.35 µm, 11(a)10 µm/10 µm, and 11(b)1.2 µm/0.35 µm. Here we compare the effect of irradiation on gain variation (δβ=βpost-βpre) with different device sizes, which is later verified by quantitative comparison (Figure 12). Thus, the conclusion is that the gain variation (δβ) becomes smaller with increasing channel length and decreasing its width.

  1. Comment: Page 13, line 285-287: Estimation method should be described in detail citing references.

Response:

We have cited related reference about mid-gap voltage method in the proper place of the revised manuscript.

  1. Comment: Page 12-14.

  Although authors discuss the effect of device geometry in terms of electric field generated in BOX layer, I think the effect can be simply understand based on negatively increased midgap or flat band voltage, rather than parasitic BJT. Surface potential bending due to the voltage gives rise to increased electric field in the gate oxide and BOX layer. Densities of positive charges accumulated in them and interface traps formed at the gate and BOX/Si are increased. It is inherent dimensional effect, since drain current of MOSFET ideally proportional to W/L. Accordingly, estimated mid gap current and voltage are also affected by W/L.

Response:

The threshold voltage shift of irradiated device is caused by the oxide-trap charges and the interface-trap charges. At low irradiation doses, the oxide-trap charges play the main role. Through the I-V transfer characteristic curve of the device, we calculate the mid-gap voltage shift before and after irradiation (∆Vmg), from which we can obtain the variation of the oxide-trap charges density  (∆Not), and then calculate the variation of the interface-trap charges density(∆Nit). It can be seen from the formula that the shift of threshold voltage is related to the oxide charges density, which excludes the factor of device size.

∆Vth = ∆Vot +∆Vit =q(∆Nit −∆Not)/Cox 

The gain of parasitic BJT is influenced by the different charges density in the BOX layer when the device of different size is irradiated. For short channel devices, the oxide-trap charges density generated by irradiation in BOX layer is higher. Moreover, the recombination current in the base area of the short-channel device accounts for a smaller proportion of the base area current, therefore the effect of interface-trap charges generated by irradiation is weaker than that of the long-channel device. The above two reasons lead to the weaker increase of the parasitic BJT gain by irradiation for the short-channel device. For wide-channel devices, the oxide-trap charges density generated by irradiation in the BOX is larger, which is due to the larger electric field when the PG bias of wide-channel devices is caused by the current crowding effect.

In addition, in order to keep the front gate channel of the device from being in inverse state, a voltage of -1V is added to the front gate when the gain of parasitic BJT is measured.

Round 3

Reviewer 2 Report

In general, resubmitted manuscript is of importance to be accepted for publication.